

# Statistical modelling of air quality impacts from 1500 individual forest fires in NSW, Australia

Michael A. Storey[1,2], Owen F. Price[1,2]

[1]NSW Bushfire Risk Management Research Hub, Wollongong, NSW 2522, Australia

5   [2]Department of Earth, Atmospheric and Life Sciences, University of Wollongong, NSW 2522, Australia

*Correspondence to*: Michael A. Storey (mstorey@uow.edu.au)






**Abstract.** Wildfires and controlled hazard reduction burns produce smoke that contains pollutants including particulate matter. Particulate matter less than 2.5 μm in diameter ($PM_{2.5}$) is harmful to human health, potentially causing cardiovascular and respiratory issues that can lead to premature deaths. $PM_{2.5}$ levels depend on environmental conditions, fire behaviour and smoke dispersal patterns. It is important for forest-fire management agencies to understand and predict $PM_{2.5}$ levels associated with a particular fire, so that pollution warnings can be sent to communities and/or hazard reduction burns can be timed to

avoid the worst conditions for $PM_{2.5}$ pollution.

We modelled $PM_{2.5}$, measured at air quality stations in NSW Australia, from 1500 historical individual fires as a function of fire and weather variables. Using VIIRS satellite hotspots, we identified days where one fire was burning within 150 km of one of 48 air quality station. We extracted ERA5 gridded weather data and fire area estimates from the hotspots for our modelling. We created random forest models for afternoon, night and morning $PM_{2.5}$ to understand drivers of and predict

$PM_{2.5}$.

Fire area and boundary layer height were important predictors across the models, with temperature, wind speed and relative humidity also important. There was a strong increase in $PM_{2.5}$ modelled with decreasing distance, with a sharp increase when the fire was within 15 km. The models improve understanding of drivers of $PM_{2.5}$ from individual fires and demonstrate a promising approach to $PM_{2.5}$ model development. However, although the models predicted well overall, there were several

large under-predictions of $PM_{2.5}$ that mean further model development would be required for the models to be deployed operationally.







## 1 Introduction


Smoke from forest fires produces pollutants harmful to human health, which have been estimated to cause ~300,000 deaths per year globally (Johnston et al., 2012). Particulates smaller than 2.5 µm ($PM_{2.5}$) are of particular concern (Haikerwal et al., 2016; Reid et al., 2016) and are criteria pollutants in the regulatory systems for air quality, for example, in the USA National Ambient Air Quality Standards and the Australian National Environment Protection (Ambient Air Quality) Measure.

Hazard reduction burns (HRB; a.k.a prescribed or planned burns) and wildfires can both produce high levels of $PM_{2.5}$. The impact of wildfire-produced $PM_{2.5}$ on populations, including hospitalisations and premature deaths, varies yearly and spatially depending on wildfire occurrence (Matz et al., 2020; Jaffe et al., 2008), which is driven by droughts, high temperatures and strong winds. Health costs associated with the 2019-2020 wildfires in Australia was estimated to be around 2 billion dollars (Johnston et al., 2021). This fire season was associated with massive burnt area, including over 5 million ha burnt in NSW

alone (Filkov et al., 2020). While wildfire ignitions and sizes are unpredictable, HRBs are planned, controlled fires conducted to reduce fuel and reduce the spread and intensity of future wildfires. There have been notable instances when HRBs caused poor air quality over large cities (Broome et al., 2016; He et al., 2016; Miller et al., 2019). Large areas of land can also be burnt under HRBs, for example, in Western Australia, 7 % of the forest is treated via HRBs each year (Bradshaw et al., 2018), while in Georgia, USA, 3-4 m ha is treated each year (Zeng et al., 2008). HRBs also typically occur closer to population centres

(Price and Bradstock, 2013) and burn under calm, still weather conditions that may be more conducive to high pollution levels (Di Virgilio et al., 2018a). Borchers-Arriagada et al. (2021) found, by comparing population weighted $PM_{2.5}$ exposure on days dominated by HRBs and wildfires, that HRBs in NSW Australia impose four-times higher health costs per hectare burnt than do wildfires. This may result from differences in fuel consumption rates (Price et al., in press), plume behaviour and/or weather. We need to better tools to help understand $PM_{2.5}$ dispersal and impacts from individual fires. Improving the tools available to

forest fire management agencies would improve pollution warnings and indicate changes that could be made to HRB strategies to reduce community $PM_{2.5}$ exposure, e.g. identifying low pollution risk days to conduct HRBs. Attributes of an individual fire that could affect their $PM_{2.5}$ output are their size, rate of heat and smoke production, proximity to human populations, and weather conditions including temperature, humidity, wind speed, wind direction, atmospheric stability and differences in weather between the HRB location and the $PM_{2.5}$ monitor location. There is some knowledge about the influence of weather

on pollution, but this has been investigated at a larger scale than individual fires. For example, it is known that days with HRBs are likely to have poorer air quality in Sydney when there is cool, stable conditions with light westerly winds (Di Virgilio et al., 2018a), while poor air quality, as measured by ozone levels, tends to occur with a high pressure system to the east of Sydney with light north-westerly winds and a sea-breeze (Hart et al., 2006).

There are a variety of ways to improve our understanding $PM_{2.5}$ from individual fires. Atmospheric dispersion models can

predict the spread of particulates from fires based on modelled atmospheric dynamics and are routinely used in many countries to guide burning operations and community warnings for HRBs. However, while evaluations of such systems are rare, existing



evaluations indicate a poor to moderate agreement between predictions and observations (Yao et al., 2014; Saide et al., 2015), possibly because local effects of HRBs are poorly captured by the models.

An alternative method is to relate air quality observations directly to real fires to calculate how far the smoke impact is likely
to spread and under what conditions. This has been done using monitors stationed up to ~10 km from HRBs (Pearce et al., 2012; Price and Forehead, 2021). Pearce et al. (2012) made 684 24-hour observations of $PM_{2.5}$ by placing monitors around 55 forest HRBs. They found that $PM_{2.5}$ concentrations fell to near-background levels within 3 km of the fire perimeters. Price and Forehead (2021) made 5445 hourly observations of $PM_{2.5}$ with a combination of fixed and mobile monitors around 18 forest HRBs. They also found that $PM_{2.5}$ concentrations had largely fallen to background levels by 3 km but that this depended on
weather conditions. One of the burns caused poor air quality at monitors more than 50 km away. These studies captured the local effects of the HRBs, but did not explain why HRBs occasionally cause impacts much further away.

Deploying air quality monitors to wildfires is difficult due to the large size of wildfires, unpredictable ignition and spread and the safety risks of working near an active wildfire. However, large permanent air quality monitoring systems can be used to gather $PM_{2.5}$ data for wildfires and HRBs, for example to NSW Air Quality Monitoring Network. Here, we use historical fire
and air quality data to identify all of the occasions when a single fire was burning within 150 km of a monitor in the NSW Air Quality Monitoring Network from 2012 to 2021, and develop random forest models of $PM_{2.5}$ concentrations at individual monitors as a function of fire area, distance and weather conditions. Our aims were:

1) Improve understanding of the fire and weather conditions that influence smoke dispersal and $PM_{2.5}$ levels.
2) Develop a predictive model of $PM_{2.5}$ output from forest individual fires, as a complement physical models, to
improve warnings.
3) Make inferences about potential changes in HRB burning protocols that could reduce $PM_{2.5}$ impacts.

## 2. Methods

### 2.1 Fire Data

Our study period was from February 2012 to September 2021 because this was when one of our fire data sets was available
(see below). For the study period, we created a spatial dataset of forest fires that were actively burning within 150 km of air quality monitoring stations (AQS) maintained by the NSW Department of Planning, Industry and Environment (DPIE) (Fig. 1). We assigned attributes of fire location, fire type (Hazard Reduction Burn (HRB) or Wildfire (WF)), date of fire activity and AQS name and location. Each fire had at least one active date and most burnt on several days. As a fire could be within a 150 km buffer of multiple AQS, there was a separate row in our data for each fire and AQS combination. For our modelling,
we used only cases where, for each AQS and day, only one fire was active within 150 km of the AQS. We did not analyse cases where multiple fires were burning on the same day near the same AQS as it was unclear which fire produced the smoke that reached the AQS.

We relied on two data sources to identify fire locations, type and active dates: NSW fire history GIS polygons (NPWS, 2021), maintained by NSW National Parks and Wildlife Service (NPWS), and VIIRS SNPP hotspots, downloaded from NASA's Fire



Information for Resource Management System (Schroeder et al., 2014; Nasa, 2021). VIIRS SNPP, which refers to the Visible

Infrared Imaging Radiometer Suite - Suomi National Polar-orbiting Partnership, hotspots were available beginning 20 January

2012.

The fire history dataset is a spatial polygon dataset of the final burnt area of fires across NSW, which has attributes of fire

identity (name and number), fire type (HRB or WF) and start and end dates. We did not rely solely on the fire history to identify

fire locations and dates because initial inspection suggested some issues for our analysis. These included fires identifiable from

VIIRS hotspots/images that were missing from the fire history; occasional errors in start and end date recording; the final fire

polygon being the combination of separate fires that eventually merged; and the data identifying only fire start and end date,

not whether a fire was actively burning on each day between those dates (e.g. fires may have extinguished then reignited on

different days). Also, the data did not capture daily fire progression only final boundary, meaning the location of fire activity

on the first day, perhaps a few hectares, is not well represented by the final fire polygon, perhaps tens or hundreds of thousands

of hectares, which is particularly an issue for WF.

We employed a process to map active fire dates and locations from clusters of VIIRS SNPP hotspots. We used VIIRS SNPP

hotspots (from here "hotspots") instead of MODIS as VIIRS are higher resolution (at nadir, 375 m vs. 1km for MODIS), thus

can detect more hotspots per fire than MODIS, which reduces the chance that an active fire is missed (Schroeder et al., 2014).

The process to create hotspot clusters for each day for each AQS was to:

1.    Extract all hotspots within 150 km of the AQS.
2.    Remove hotspots that were not in forest by removing hotspots with low foliage projective cover score (Gill et al.,
      2017). We removed hotspots with foliage projective cover < 125.
3.    Buffer each hotspot by 2.5 km and dissolve overlapping buffers into a single polygon, thus creating hotspot cluster
polygons (Fig. 2).
4.    Record the number and area of day and night hotspots in each cluster. Area of an individual hotspot was the VIIRS
      pixel width by height (i.e. "Scan" and "Track" attributes), which varies with scan angle, and the total area for a
      cluster was the sum of the individual hotspot areas.
5.    Remove clusters that did not have at least five day or night hotspots. This was our minimum threshold for fire
activity, as we wanted to exclude small fires such as burning heaps on farmland that can be detected by VIIRS. We
      also tested three as a minimum threshold, which produced similar but less accurate models.
6.    Repeat process for each combination of date and AQS.

Where a fire identified from the above process (a "VIIRS fire") intersected a NPWS fire history polygon between its start and

end date, we assigned the fire name, number and type (HRB or WF) to the VIIRS fire. If multiple VIIRS fires intersected the

same fire history polygon, we merged them into a single fire with the same attributes for analysis. If a VIIRS fire intersected

multiple fire history polygons, we assigned the attributes from the fire history polygon with the largest overlapping area. Fire

history data were excluded from analysis if either the start or end date was still missing after all of the checks, or they had no

matching intersecting VIIRS hotspots. If a VIIRS fire did not intersect a fire history polygon, we assigned the fire type based

on date: from November to January (inclusive) were WF, all other months were HRB. For each fire identified we added

attributes of distance and direction from AQS to fire centroid (Fig. 2), i.e. the arithmetic mean of the hotspot coordinates, with

a separate row for each fire and AQS combination (within 150 km).



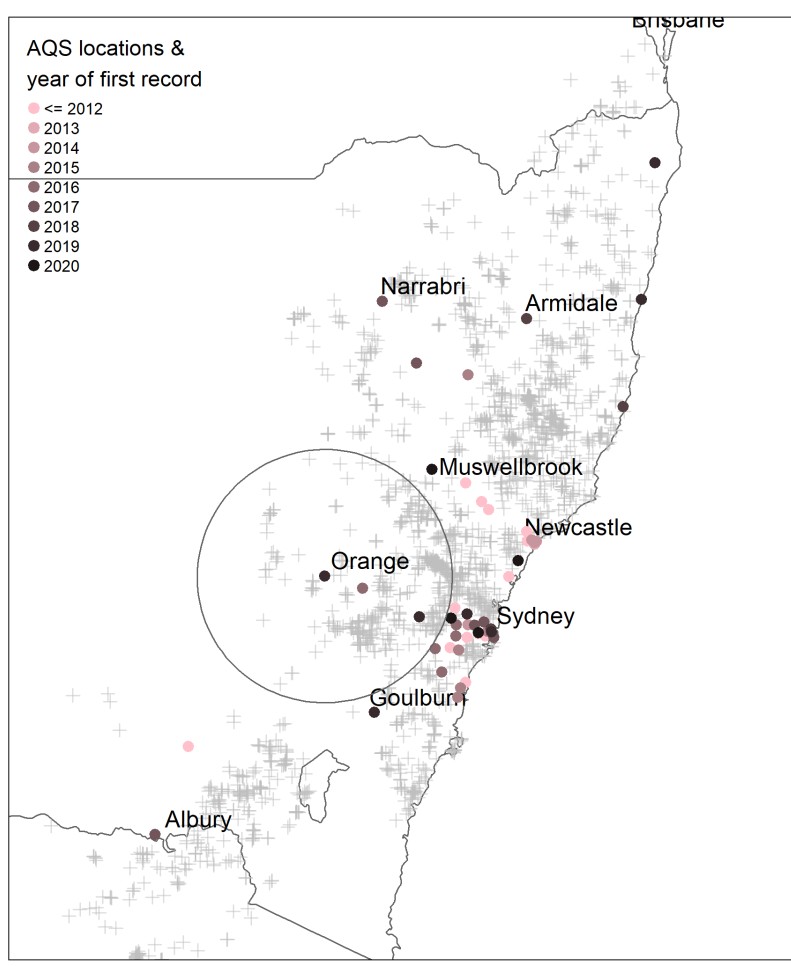

**Figure 1: Map of study area (New South Wales, Australia) showing air quality monitoring stations (AQS, n=48), coloured by year of first PM$_{2.5}$ record. Grey crosses are the locations of all fires used in analysis, with one cross per-fire per-day. 150 km buffer shown around Orange AQS as example (all AQS had 150 km buffers for analysis).**





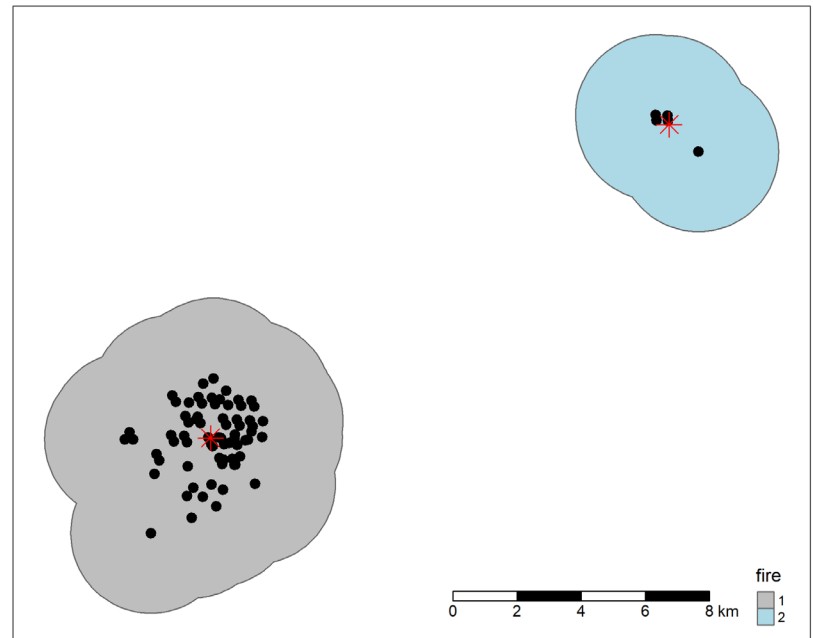

**Figure 2: Example of creating clusters from VIIRS hotspots. Black are VIIRS SNPP hotspots, red asterisk is fire centroid, i.e. the arithmetic mean of the hotspot coordinates. The image has two separate fires. Each hotspot is buffered by 2.5 km, all overlapping buffers merged, and hotspots assigned to each separate buffer. Two separate fires are created here because of distinct fires where buffers don't overlap, i.e. greater than 2 buffer widths (5 km) apart. Fire 1 (grey) has > 50 hotspots, fire 2 (blue) has 5 hotspots. Fire area was calculated from hotspots pixel size, not buffer size.**

## 2.2 PM$_{2.5}$ Data

We modelled PM$_{2.5}$, particulate matter < 2.5 µm diameter as micrograms per cubic metre of air (µgm$^{-3}$), as a function of several environmental predictors. We downloaded all available PM$_{2.5}$ data (hourly averages) from the NSW DPIE for the period 2012 – 2021, which comprised of 48 AQS. Data are available free online at https://data.airquality.nsw.gov.au/docs/index.html. We calculated mean PM$_{2.5}$ for each AQS for three six-hour time-periods:

1.  **Afternoon**: 1400 to 1900 AEST inclusive. This period covered peak burning conditions in the afternoon and after sunset, although sunset and fire ignition times varied.
2.  **Night:** 2100 to 0200 AEST inclusive. Covered the night period starting on the same day as the fire.
3.  **Morning:** 0500 to 1000 AEST inclusive, next day after fire day. Captured early next morning conditions after the main periods of fire activity are likely to have ended, although some fires may have burnt through the night and smoke may still have lingered.

We chose these times to represent different periods in the daily cycle that may have distinct smoke, weather and fire behaviour characteristics. All fires identified in the hotspot analysis were matched to AQS summary PM$_{2.5}$ for active days, when the fire was within 150 km. Not all AQS had records for all years, as some were not operational until later in the study period (Fig. 1).

### 2.3 Predictor variables

We sampled hourly weather variables at each AQS and each fire centroid from ERA5 weather grids, which is an atmospheric reanalysis product with multiple weather variables and atmospheric levels available at 30 km spatial and hourly temporal



resolution (Hersbach, 2021b, a) (Table 1). We calculated the mean weather values for both surface and upper atmospheric conditions (Table 1) for the afternoon, night and morning periods as described for $PM_{2.5}$. We calculated additional variables describing the spatial relationship between the fire and each AQS. We used the AQS to fire direction and wind direction to

calculate the percent of time-period where surface wind was blowing directly to the AQS, with directly meaning ±22.5 degrees of the AQS to fire bearing. We also calculated the sum of the hotspot day and night fire area as a predictor. We included the month of the active fire date in the modelling to account for account for seasonal variation in background $PM_{2.5.}$ Month was represented a cyclic variable, where the sine and cosine of the month (1-12) were both included in modelling. We included the latitude and longitude of the AQS to account for spatial dependence, and fire type as a factor variable to account for differences

not captured by the weather/fire area variables. We also experimented with making separate models for each fire type (HRB model and WF model) for each time period, but found that resulting accuracy statistics on the training and test sets were similar, so instead just used one model for each time period including fire type as a factor variable.







**Table 1: Predictor variables used for random forest modelling. Letters mean that for the random forest modelling, weather variables were sampled at the fire (F), at the AQS (A) or both (FA). MSLP and wind speed (850 hpa) at the AQS were excluded due to being highly correlated with the same variable at the fire.**

| Type | Name | Units | Details |
|------|------|-------|---------|
| ERA 5 weather | **PBLH** – Planetary boundary Layer Height (FA) | metres | Mean height of planetary boundary layer from surface, from ERA 5 grids. |
| | **MSLP** – Mean Sea Level Pressure (F) | hectopascal | Mean sea level pressure of atmosphere on surface per unit area from ERA 5 grids. |
| | **WS** – Wind speed (FA) | km h$^{-1}$ | Mean wind speed 10 m above surface calculated from U and V ERA5 wind component variables. |
| | **RH –** Relative humidity (FA) | % | Mean relative humidity calculated from temperature and dew-point ERA5 variables. |
| | **Temperature** (FA) | Celsius | Mean temperature 2 m above surface sampled from ERA5 grids. |
| | **WS 850 hpa –** Wind speed at 850 hpa (F) | km h$^{-1}$ | Mean wind speed at 850 hpa calculated from U and V ERA5 pressure-levels wind component variables. |
| | **Direct wind** (FA) | % | Percent of hours during period where 10 m wind was blowing directly toward AQS, i.e. within 22.5 degree arc either side. |
| Fire | **Fire area** | hectares | Hectares calculated from VIIRS hotspot pixel size (scan*track). Sum of day and night hectares. |
| | **Fire Type** | WF or HRB | Wildfire or hazard reduction burn |
| Temporal | **Month** | sine, cosine | Month included account for seasonal variation in background PM$_{2.5}$. Included as cyclic variable: cosine and sine of integer month as separate variables. |
| Geographic | **Distance** | km | Kilometres from fire centroid, i.e. geometric centre of hotspots in a cluster, to the AQS |
| | **AQS coordinates** | Latitude, Longitude | Coordinates of air quality monitoring stations, to account for spatial dependence. |



## 2.4 Random Forests Modelling

Our data consisted of three tables (afternoon, night and morning data tables) each of 11187 rows with unique combinations of fire, AQS and date. We refer to these combinations as "fire-days". For each fire, there could be multiple active dates and each fire could be within 150 km of multiple AQS. There were 48 AQS, 1429 dates and 1546 unique fire IDs in our dataset. There were 1883 different combinations of fire and date consisting of 727 fire-days that had VIIRS hotspots and a fire history record, and 1156 fire-days that had only VIIRS hotspots. The fire-days from VIIRS hotspots only were on average smaller than the

fire-days with a matching NPWS fire history record (304 ha vs 1176 ha respectively). 1182 fire-days were from HRBs (mean area = 346 ha) and 701 were from WFs (mean area = 1137 ha). Each fire was observed at a minimum of one AQS, with a mean of six AQS and maximum of 35 AQS associated with a single fire.

We trained a random forest model using the "ranger" package in R (Wright and Ziegler, 2017). We split our dataset into training (75 %) and test (25 %) sets for analysis, stratified by fire type so that an even number of HRBs and WFs appeared

within each of the sets. We trained the models using the training set data and used out-of-bag (OOB) predictions vs observations for model accuracy checks. We used the model to predict to the test set to calculate test set accuracy statistics. Our accuracy statistics were the correlation coefficient ($r$), normalised mean error (NME) and normalised mean bias (NMB), as recommended in Emery et al. (2017) for assessing model performance. We ran three different models, one for each analysis period: 1) afternoon mean $PM_{2.5}$, 2) night mean $PM_{2.5}$, 3) morning mean $PM_{2.5}$. Predictor variables were the weather variables

in Table 1 sampled at both the AQS centroid and fire centroid, distance, fire area, month and AQS coordinates. As highly correlated variables can introduce bias into random forests variable importance calculations (Strobl et al., 2008), we removed variables from analysis where correlation was above 0.8: MSLP at AQS and wind speed 850 hpa at AQS were excluded, each of which we correlated with the version sampled at the fire.

We assessed variable "permutation" importance using in the ranger package. Permutation importance is derived from a process

where reduction in model accuracy on OOB predictions is calculated after randomly shuffling values for each variable, calculated for all trees and variables (Wright et al., 2016). We assessed predictor variable effects using partial dependence plots calculated in the "pdp" package in R (Greenwell, 2017), and by creating prediction plots where $PM_{2.5}$ was predicted with all variables held at mean values except two variables of interest, which were each assigned three different levels to illustrate their effects. We also conducted a short descriptive analysis, using satellite images and hourly $PM_{2.5}$ of large outliers in the

models to understand potential reasons for inaccurate predictions. This is included as Appendix A.

## 2.5 Limitations

There are several limitations to our methods that should be considered when interpreting the results. Our process to identify active fires from VIIRS hotspots excluded hotspots that were outside the 150 km AQS buffer, even if they were part of a fire that straddled the buffer edge. There may be occasions where smoke from hotspots, and entire fires, from > 150 km reached





an AQS and influenced PM$_{2.5}$, e.g. large WFs during the 2019-20 "Black Summer". The effect of such fires was not captured in our methods.

We set a minimum fire activity threshold of five hotspots (day or night). This may mean that days recorded as having only one fire may have had other smaller fires in the area that may have produced smoke that affected PM$_{2.5}$. Relying on VIIRS had the advantage of being able to better detect when a fire was active, but our process may not have captured all fires on any given

day due to cloud cover impeding VIIRS hotspot detection. This may be a form of bias in our analysis as the cloudiest days were selected against. Additionally, VIIRS SNPP hotspots are acquired early afternoon and early morning, meaning that total burnt area on a day is not measured, only active area at the time of VIIRs acquisitions. Fire area, or the number of fires, may have been underestimated if cloud was impeding hotspot detection. Our decision to analyse only days with one fire, to better understand distance and direction variables, means that there is a selection bias against the most active fire days (i.e. days with

multiple fires). This may include the worst WF days, where multiple fires are more likely to ignite, particularly during 2019-20. For days that are most suitable for HRBs, authorities are more likely to ignite multiple HRBs. Such days, which could include the worst pollution events, are not included in our analysis, but they are currently part of a separate research project.

## 3. Results

### 3.1 Variable summaries

Plots of the distribution of PM$_{2.5}$ and predictor variables are shown in Figure 3. PM$_{2.5}$ was skewed toward low values (afternoon, night, morning mean = 8.1, 10.8, 10 µgm$^{-3}$), with occasional very smoky periods (afternoon, night, morning maximum = 294.2, 394.8, 506.2 µgm$^{-3}$). Most fires were between 75 and 150 km from the monitoring stations, and only 20 % of fires had their closest AQS within 50 km. Fire area derived from VIIRS hotspots was heavily skewed toward lower values (mean = 640 ha, 90$^{th}$centile = 993 ha). Maximum fire area was 56178 ha, < 1 % of fires (all WF) were over 10000 ha and 90

% were less than 1000 ha.

Afternoon conditions were generally hotter, less humid and had higher PBLH at both fire and AQS locations than were nights and mornings. Between WF and HRB, WF afternoons were hotter, drier and had higher PBLH (Fig. 3). MSLP was similar between afternoon, night and morning, but skewed lower for WF compared to HRB. The wind direction variables were clustered around zero, indicating that most of the time wind at the fire and at the AQS was not moving smoke directly from

the fire to the AQS (Fig. 3). For example, only 5 % of rows in the afternoon data indicated that wind sampled at the AQS was coming directly from the fire for at least 3 of the 6 hours (50%). For wind sampled at the fire during the afternoon, this figure was 11 %.





**Figure 3: Distribution of PM$_{2.5}$ and predictor variables used in random forest modelling, excluding latitude, longitude, fire type and month. Distance and fire area are daily variables, so are identical for afternoon, night and morning model datasets. Distributions are from unique fire-day-AQS combinations. af=afternoon, ni=night, mo=morning. AQS=Air Quality Station, RH=Relative Humidity, WS=Wind Speed, PBLH =Planetary Boundary Layer Height, MSLP=Mean Sea Level Pressure.**

### 3.2 Highest PM$_{2.5}$ days

Figure 4 shows the 20 highest mean PM$_{2.5}$ values for each six-hour time-period for HRBs and WFs. The top PM$_{2.5}$ values were much greater for WFs than for HRBs in the afternoon, night and morning (~150 to 200 µgm$^{-3}$ greater for each). >= 80 % of the top 20 PM$_{2.5}$ values for WF for afternoon, morning and night were associated with the 2019-2020 wildfires in NSW, many with the Gosper's Mountain wildfire in the Blue Mountains (Boer et al., 2020).





The top seven afternoon peaks for WF were > 100 µgm⁻³ (max= 294 µgm⁻³) but only two of the afternoon HRB peaks were > 100 µgm⁻³. In the night and morning, there were fewer values > 100 µgm⁻³, but larger maximums were record for HRB and

WF for each period (compared to the afternoon). For each rank position, WF values were greater than HRB values, except in the night model where from positions 3 to 20, the HRB values were higher. More information, including satellite images, weather plots and descriptions, on the conditions surrounding worst $PM_{2.5}$ events for each time period for HRBs and WFs is included in Appendix A.

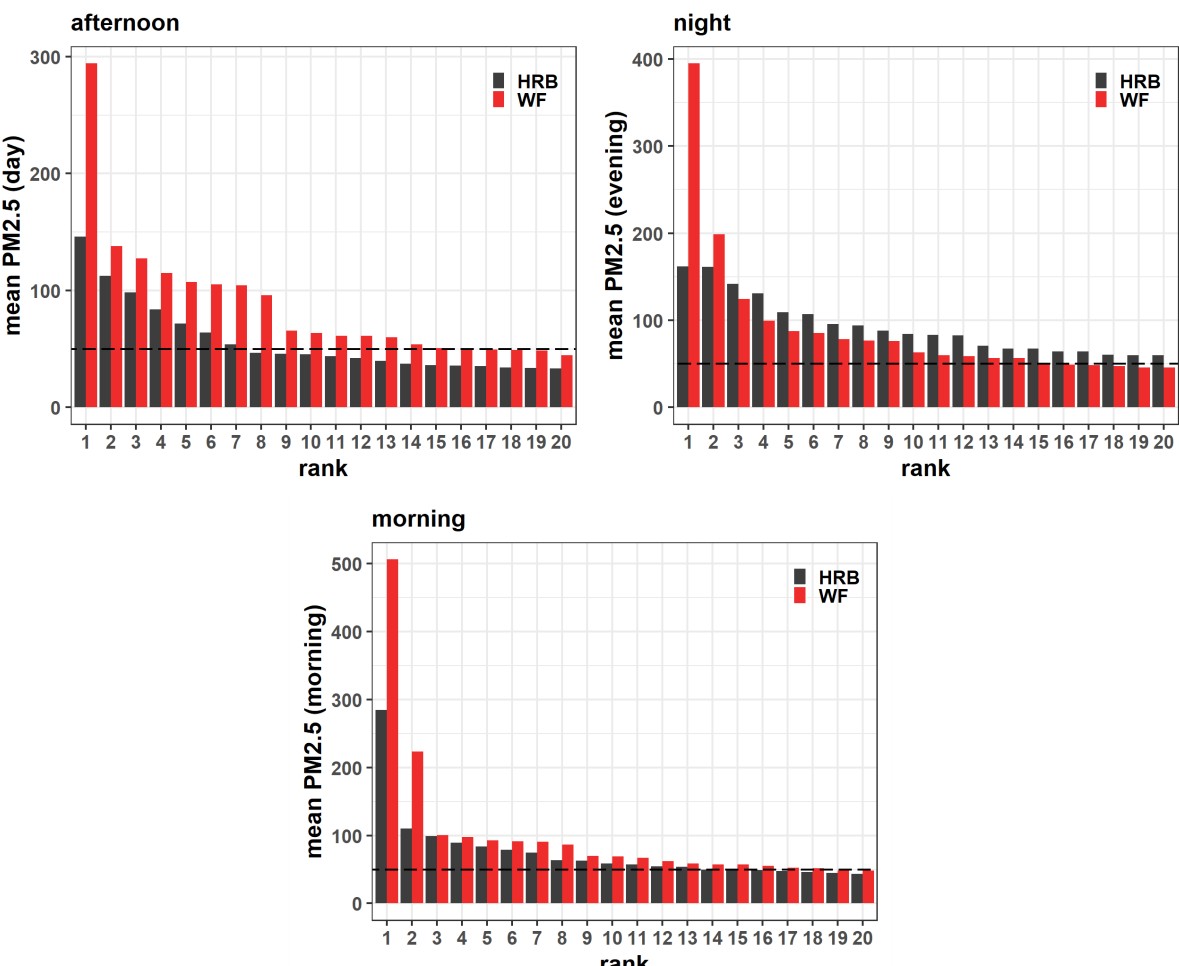


**Figure 4: Highest mean $PM_{2.5}$ values for each six-hour time-period for HRBs and WFs. Only the top values for each date is shown. This means if the second highest value was over 100 for a particular date at another AQS, it is not included here. Dashed line indicates 50 µgm⁻³ for reference between the three plots. Note that our data only includes situations with one fire within 150 km of AQS for a particular date.**





### 3.3 Model results

Fire area, PBLH (fire and AQS), temperature and RH at the fire were among the most important variables in the three models (Fig. 5). Some variables were among the most important in only one or two of the models: wind speed at the fire was the second most important in the night model, but eighth most important in the afternoon model. The direct wind variables, distance to fire, AQS coordinates, MSLP, month and fire type were all of moderate to lower permutation importance in each model.

Partial plots (Fig. 6) indicated that for all models, there was a sharp increase in predicted $PM_{2.5}$ when the AQS was below ~ 20 km from a fire, with the morning model displaying the sharpest rise in $PM_{2.5}$ as the distance decreased. This effect is despite distance being of middle to lower permutation importance (Fig. 5). Partial plots indicated $PM_{2.5}$ increased as fire area increased, particularly in the 0 to 2500 ha range, which is where most training observations were situated (Fig. 3). There was a very large $PM_{2.5}$ increase above 10000 ha in the morning model, although there is uncertainty here due to small number of training observations > 10000 ha (Fig. 3). The shape of the PBLH effect differed for each model between the fire PBLH and AQS PBLH. At the AQS, there was a strong negative effect of PBLH (lower PBLH = higher $PM_{2.5}$), particularly in the night and morning models. At the fire, each model had peak $PM_{2.5}$ at low and high values of PBLH. For the night and morning models, $PM_{2.5}$ peaked when fire PBLH was < ~ 200 m, with a smaller rise > ~ 800 m. For the afternoon model, the largest peak was when fire PBLH was high (> ~1500 m), with a smaller rise when < ~ 500 m. RH at the fire had an almost threshold effect in the morning and night models, where predicted $PM_{2.5}$ below ~ 50 % RH was much higher than when RH was above 50 %, particularly for the morning model. For wind speed, effects varied between the fire and AQS, and time period: lower wind speed at the AQS was associated with higher $PM_{2.5}$ in all models, but at the fire, low and high (particularly for the night model) wind speeds were associated with higher $PM_{2.5}$.

We calculated model accuracy statistics for the training set (OOB predictions) and the independent test set, and for HRB and WF subsets of each set. From the combined statistics, all correlations between predictions and observations ($r$) for training and test sets were > 0.7, except for test set predictions for the night model where $r$ was 0.58 (Table 1, Fig. 7). The morning model produced the higher $r$ on the training set and test sets (0.8 and 0.79). For the statistics by fire type, $r$ was generally higher for WF than for HRB. For WF, $r$ was >= 0.8 on the training set (max. = 0.86 for morning) and on the test set $r$ was >= 0.75 expect for the night model, where $r$ was 0.52. For HRBs, $r$ was 0.58 to 0.63 on the training set, and 0.66 to 0.69 on the test set. NME for all combinations of training/test set and fire type ranged between 32 % and 40 %, but the best result was for the WF subset from the afternoon model, where NME was 33 % and 32.3 % for the training and test sets. The NBE error indicated that in generally there was a slight over-prediction bias that ranged from ~1 % to ~6 %, although the afternoon model had slight under-prediction bias on the test set.

The models had large under-predictions for the largest $PM_{2.5}$ values, but also some large over predictions (Fig. 7). NBE calculated on data that included only where observed $PM_{2.5}$ was >= 20 was -32.5 % (training) and -34.2 % (test) for the afternoon model, -57.8 % and -51.9 % in the night model, and -24.8 % and -24.1 % in the morning model, indicating under prediction bias for the larger $PM_{2.5}$ values. For predictions to the test set, in the afternoon model there were 13 observations





that were under-predicted by at least 30 $\mu gm^{-3}$, 8 of which were WFs (6 of the 8 were from 2019-2020 bushfires). The maximum over-prediction was by 30 $\mu gm^{-3}$. For the night model, there were 20 occasions where the model under-predicted to the test set

by at least 30 $\mu gm^{-3}$ (8 were WF and 12 were HRB), with the biggest under-prediction by 378 $\mu gm^{-3}$ for a 2019-2020 WF (see Appendix A Fig. A1). The maximum over-prediction was by 57 $\mu gm^{-3}$. The morning model had 15 under-predictions on the test set by at least 30 $\mu gm^{-3}$, with the largest under-prediction by 74 $\mu gm^{-3}$. The were 4 over-predictions by at least 30 $\mu gm^{-3}$, with a maximum over-prediction by of 158 $\mu gm^{-3}$.

We explore the influence of distance and some selected variables with a series of prediction plots (Fig. 8). PM$_{2.5}$ was predicted

to increase substantially with decreasing distance within the first 15 km of the fire in all combinations fire area, PBLH, RH and temperature in Figure 8. Beyond ~ 30 km there was minimal to no effect of distance, except in the morning model with large fire area and low PBLH at the AQS (Fig. 8a). Note that these conditions were rare in the training data (Fig. 3). The effect of temperature at the fire differed between models, such that as temperate increased from 10 to 25 C, PM$_{2.5}$ was predicted to decrease in the morning model but increase in the afternoon model. The plots also suggest there is generally a small difference

between predicted mean PM$_{2.5}$ for WF and HRB for each model once the other predictors including fire area are controlled for.



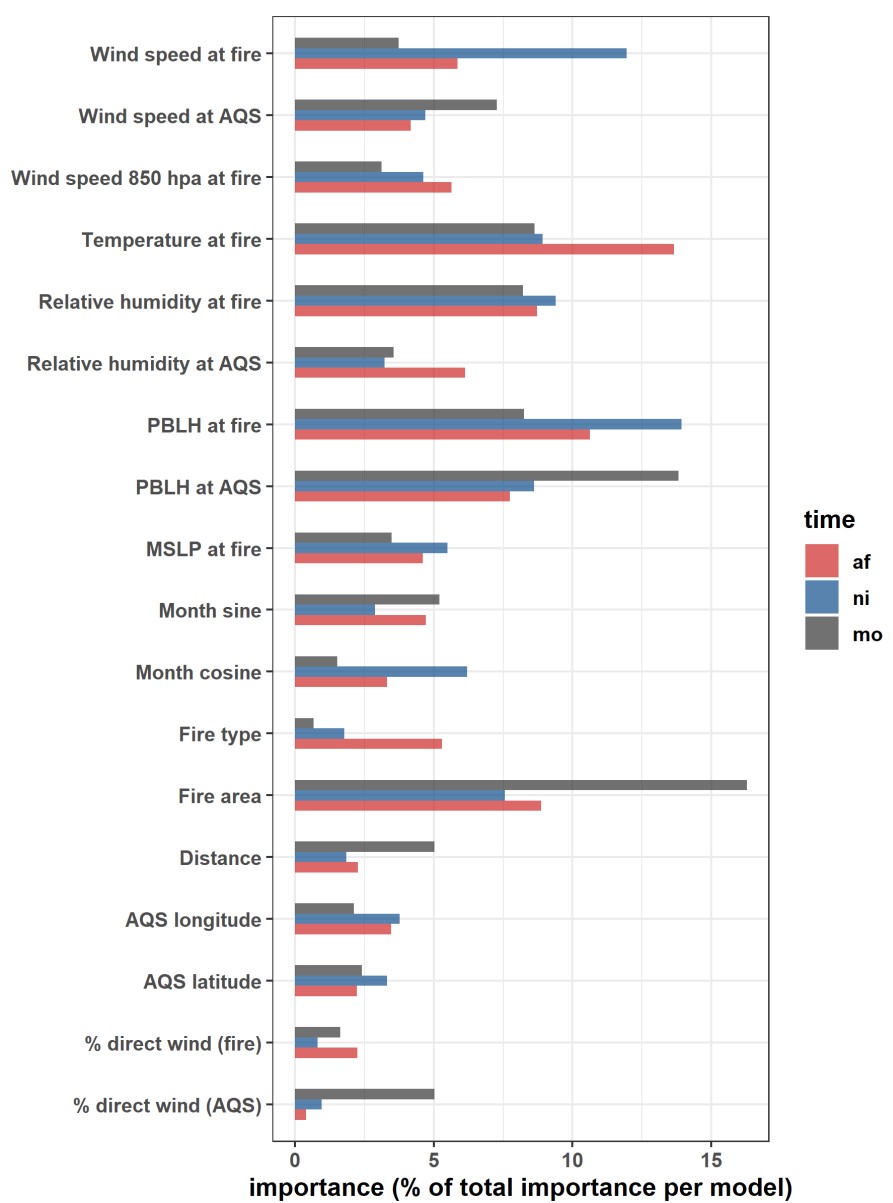

**Figure 5: Variable importance for each model. A common x scale was assigned for clarity, which is the % of the total permutation importance attributable to each individual variable (i.e. importance/sum(importance)*100).**








**Figure 6: Partial dependence plots for the afternoon (red), night (blue) and morning (black) models. Dotted parts of lines are minimum to 5th centile and 95th centile to maximum values for each predictor variable, calculated from the training data. Where dotted parts are long, this indicates a large range of values with small number of observed points for model training.**







**Table 1: Accuracy statistics from random forest modelling for training and test sets (bold in brackets). Training set predictions are on out of bag samples during model fitting, test set predictions made to independent test set. Overall statistics, along with statistics on HRB and WF portions of the data are shown. $r$ = pearson correlation, NME = Normalised mean error, NBE = Normalised bias error (Emery et al., 2017).**

|  |  | $r$ | NME % | NBE % |
|---|---|---|---|---|
| **Combined** | Afternoon | **0.75** (0.75) | **35.8** (35.3) | **1.37** (-1.5) |
|  | Night | **0.71** (0.58) | **37.1** (38) | **1.65** (1.63) |
|  | Morning | **0.8** (0.79) | **37.6** (36.2) | **2.15** (4.93) |
| **HRB** | Afternoon | **0.58** (0.69) | **38.1** (37.9) | **1.76** (-0.15) |
|  | Night | **0.63** (0.66) | **37.9** (37) | **0.72** (1.1) |
|  | Morning | **0.59** (0.69) | **38.5** (36.7) | **1.8** (5.68) |
| **WF** | Afternoon | **0.8** (0.75) | **33** (32.3) | **0.92** (-3.05) |
|  | Night | **0.8** (0.52) | **35.4** (39.9) | **3.52** (2.71) |
|  | Morning | **0.86** (0.83) | **36.3** (35.4) | **2.7** (3.86) |





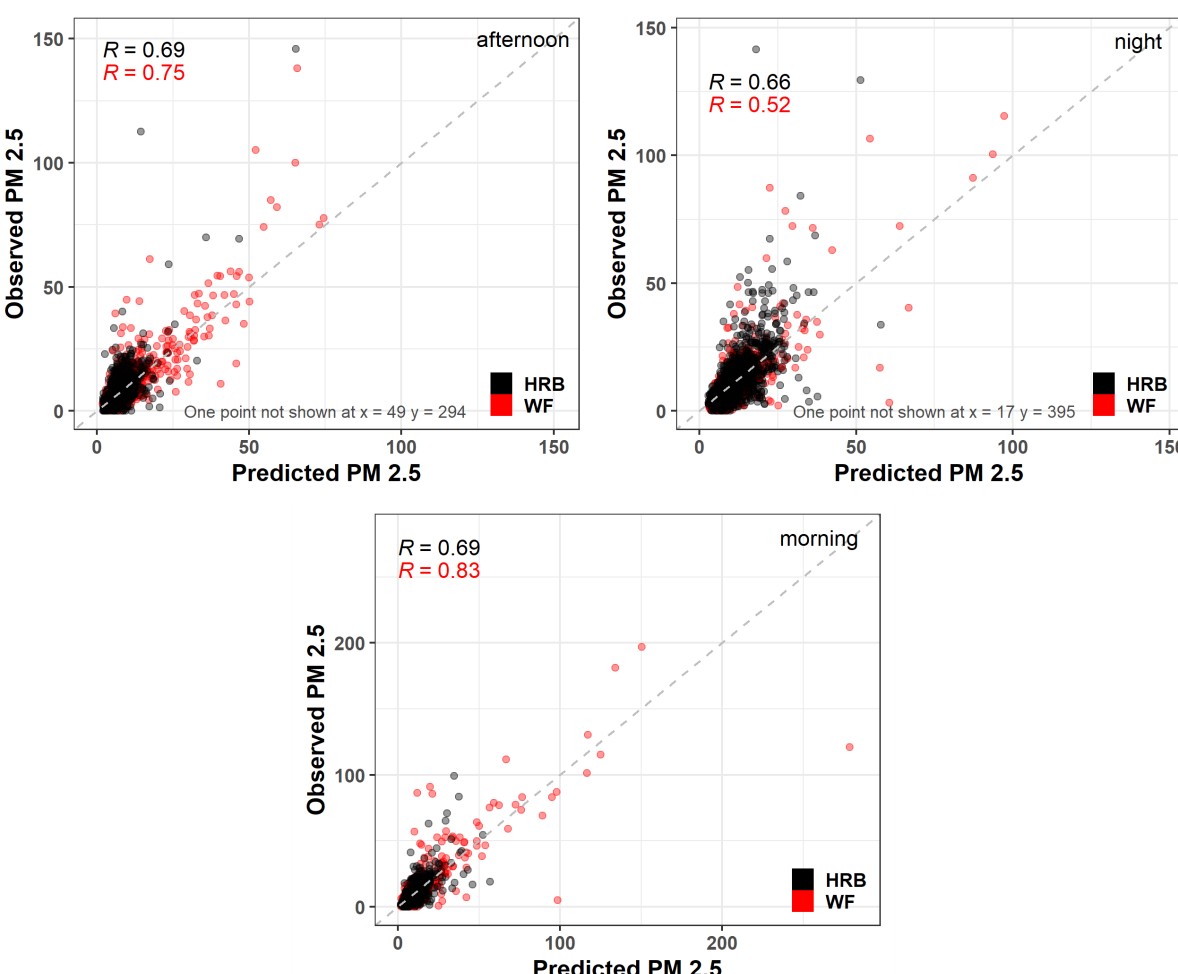

**Figure 7: Predictions of each model to test set, with points coloured by fire type. Pearson correlation of predictions to observations by fire type shown in text (*R*).**



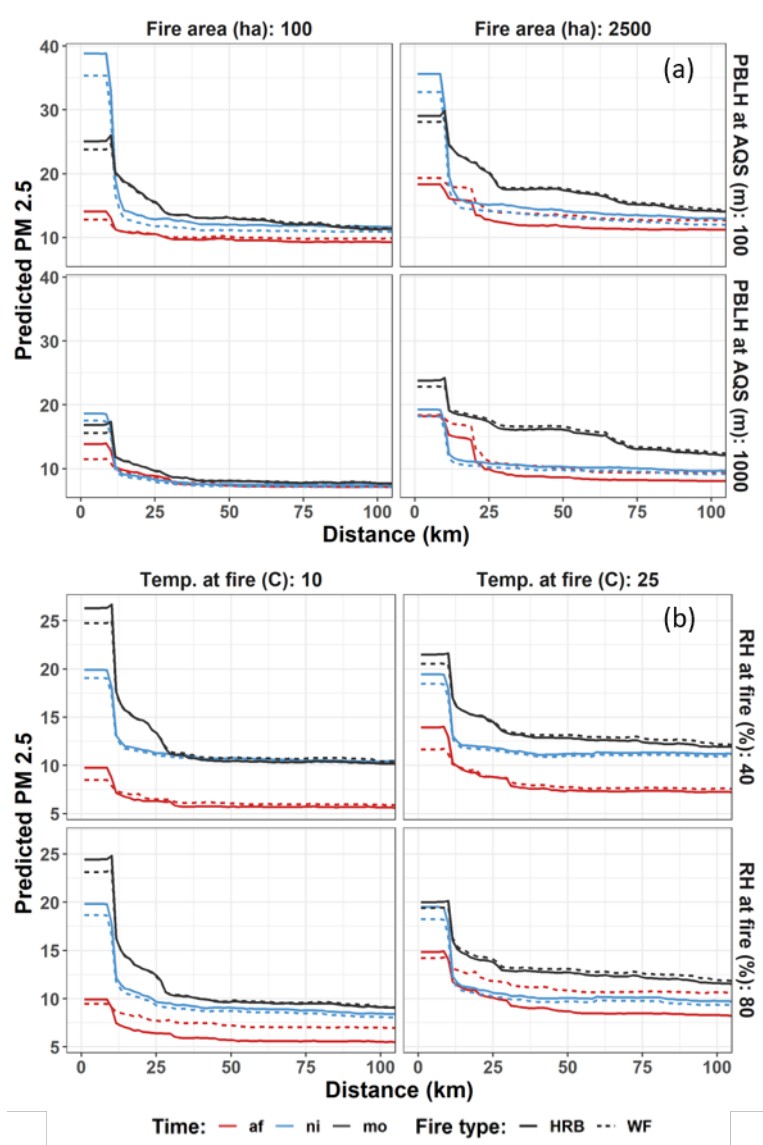

**Figure 8: Prediction effects plots of mean PM$_{2.5}$. Colours are time periods, line types are fire types and grid squares are combinations of fire area and planetary boundary layer height at the monitoring station (a), and relative humidity and temperature at the fire (b).**

## 4. Discussion

Using empirical fire and air quality monitoring station data, we identified important drivers of particulate pollution associated with individual forest fires. The results are important in the context of our first research aim, which was to ***improve understanding of the fire and weather conditions that influence smoke dispersal and*** PM$_{2.5}$ ***levels***. In our models, fire area, PBLH, relative humidity and temperature were all important drivers of PM$_{2.5}$ from individual fires. The importance of these variables at the fire or at the AQS varied between models. Distance to fire generally had low permutation importance, possibly



due to the low number of AQS in the 0 to 50 km range (Fig. 3, Fig. 6). However, partial plots and prediction plots indicated a large influence on model predictions. For example, partial and prediction plots suggested that within 15 km of a fire, $PM_{2.5}$

levels rose steeply with decreasing distance. The effect of distance > 50 km was negligible in most cases, suggesting other factors are more important drivers at such distances, although under certain conditions there could be raised $PM_{2.5}$ at long distances, such as with higher fire area and high PBLH in the morning model (Fig. 8). Note these conditions are likely associated with the worst wildfire conditions, given that high PBLH at the fire during the morning was rare and only associated with WF (Fig. 3). Based on Reisen et al. (2018), a 1000 ha prescribed burn will emit 160 tonnes of $PM_{2.5}$, enough to fill to

exceedance level a cylinder capped by a planetary boundary layer of 500m to a radius of 64 km. This means there are sufficient particulates available for a distance effect to occur should the weather conditions suit. Other authors have found similar variables to be important in modelling $PM_{2.5}$, including fire size and distance when $PM_{2.5}$ was measured within ~10 km of HRBs (Pearce et al., 2012; Price and Forehead, 2021). PBLH was also a consistent predictor of $PM_{2.5}$ levels at multiple stations in Sydney during HRB days (Di Virgilio et al., 2018b). However, studies such as these have modelled $PM_{2.5}$ over smaller

scales than we did here or did not attempt to link individual fires to $PM_{2.5}$ records. Our data included $PM_{2.5}$ measurements up to 150 km from a fire, and we built $PM_{2.5}$ models using a much larger dataset of fires and $PM_{2.5}$ records, which here were from pre-installed permanent AQS. Therefore, the results from our study are more applicable to the individual fire and $PM_{2.5}$ relationship across large geographical areas than other studies.

Our models suggest the area potentially affected by $PM_{2.5}$ from fires is larger than in Price and Forehead (2021), where raised

$PM_{2.5}$ levels were mostly modelled to be within 5 km of HRBs. Here, our models suggested raised $PM_{2.5}$ levels mostly within 15 km of a fire. Our dataset includes a larger set of fires and includes WFs, which are likely to produce smoke that travels further. In some individual cases in our raw data, fires caused high $PM_{2.5}$ levels > 100 km away (e.g. Appendix A Fig A3). Although relatively sparse, analysis using the more remote AQS network is more suited to detecting these longer-range effects than when monitors are placed only close to a fire.

Our second aim was *to **develop a predictive model of $PM_{2.5}$ output from individual forest fires, as a complement to physical models, to improve warnings***. There was some success here: *r* on the test sets indicated moderate to good agreement between predictions and observations: 0.75, 0.58 and 0.79 for the afternoon, night and morning models respectively. The models fit better on the WF portion of the training data (*r* 0.8 to 0.86) than for HRBs (*r* 0.58 to 0.63). On the test data for the afternoon and morning, correlations were higher for WF than for HRB, but lower for WF than for HRB for the night model. The generally

better results for WF suggests the models may be more applicable to WFs, e.g. for the issuance of pollution warnings due to WF smoke, rather than for assisting with HRB planning. An important consideration for using the models for prediction is their accuracy on the largest $PM_{2.5}$ observations. Events with very high $PM_{2.5}$ have the largest health impacts and are therefore most important to predict, for example to correctly issue warnings or defer HRBs due to high pollution risk. Our results suggest that, while some predictions for the largest $PM_{2.5}$ observations were relatively accurate, the models did not consistently predict

larger $PM_{2.5}$ events, so may not be suitable for as an operational prediction tool without further development.



There are several possible reasons for the biggest outliers and limited accuracy. The AQS network is relatively sparse, being concentrated in greater Sydney, making the distance between any fire and AQS usually large. The mean distance to the closest AQS to each of our fires was 87 km ($10^{th}$centile = 31 km). This may partly explain why we did not detect wind direction effects. Price et al. (2012) also did not find significant effects of wind direction effect when modelling $PM_{2.5}$ in relation to

MODIS hotspots at similarly broad scales around Sydney and Perth. In contrast, two empirical studies that did detect clear wind direction effects from HRBs, Pearce et al. (2012) and Price and Forehead (2021), placed $PM_{2.5}$ monitors close to HRBs, mostly within ~ 10 km. The large distances in our data mean smoke was subject to broader weather circulation patterns before reaching an AQS, such as shown in Appendix A. Broader circulation patterns in the Sydney basin can include westerly terrain-related drainage flows, sea breezes and their interaction (Jiang et al., 2017). The large distances and sparse network also mean

there was the low chance of any particular AQS being downwind from a fire. This is indicated by the wind direction variables being clustered around zero (i.e. smoke not blowing from fire to AQS, see Fig. 3) and in cases such as Appendix A Fig. A1, where only two from > 20 AQS detected the smoke from a WF. It may therefore be that the models were mostly optimising for non-smoke related $PM_{2.5}$, so it is not surprising that peak events are under-predicted. Our approach is promising, however, and more data capturing single fires burning near monitoring stations is likely required to produce better models. More data

could be gathered from the same AQS for another analysis in the future, or by increasing the density of $PM_{2.5}$ monitors, either through installing more permanent AQS or via a short-term project that installs a network temporary AQS in a selected fire prone area (e.g. Blue Mountains) in times of high-expected fire activity.

Some of the variables had interesting non-linear affects. For example, wind speed at the fire during the afternoon was associated with high $PM_{2.5}$ both when wind speed was < ~ 7 km h$^{-1}$ and > ~ 15 km h$^{-1}$ (Fig. 6). Such relationship are due to complex

factors. For example, it may be that low wind speeds increase $PM_{2.5,}$ because previously emitted smoke is more likely to linger, whereas high wind speeds mean that fires are more intense and produce more smoke and particulates. In other words, low wind speed increases smoke concentration at the receiver and high wind speed increases smoke production. The low wind speed effect may be more associated with HRBs, which are conducted in calm weather, and the high wind speed effect associated with WFs. Similar non-linear relationships also exist for other variables, to varying degrees, including PBLH, RH,

temperature and MSLP (Fig. 6). Some variables differed in their effects substantially between the fire and AQS. For example, afternoon PBLH at the fire showed increases in $PM_{2.5}$ at low and high levels, but at the AQS it was only low PBLH that increased $PM_{2.5}$. The PBLH effect at the fire may be similar to wind: low PBLH traps smoke but high PBLH is associated with more active fire behaviour and greater smoke production. Note that there is uncertainty about the strength and directions of the effects at the extremes of the predictor variables, given the lower proportion of observations for model training, as indicated

in Figure 6.

Our models predict only small differences between $PM_{2.5}$ depending on the fire type variable (HRB or WF), which also had low permutation importance in all three models. It is likely that the weather variables and fire area variables included in our model captured most the differences between HRBs and WFs (e.g. WF on average are larger and burn in hotter windier weather), making the fire type variable mostly redundant in the models. In this case, the models suggest that after accounting





for weather and fire size, there are no clear differences in WFs and HRBs in terms of PM$_{2.5}$ output. However, other studies have indicated that fundamental differences may exist as WFs inject smoke higher into the atmosphere and consume more fuel per hectare than HRBs (Price et al., 2022; Price et al., 2018; Volkova et al., 2014), thus WF and HRB differences need more investigation.

Our third aim was to *__make inferences about potential changes in HRB protocols that could reduce PM$_{2.5}$ impacts.__* The

models indicate the potential combinations of environmental and fire conditions where PM$_{2.5}$ is likely to be higher and fire managers must carefully consider whether to undertake HRBs due to PM$_{2.5}$ pollution risk. For example, a large HRB < 15 km from a town where PBLH < 250 m during the night and morning (at both fire and receiver site) and < 1000 m during the afternoon. When HRBs are > 50 km from a town, high PM$_{2.5}$ impact is much less likely, although certainly still possible (Appendix A). In addition, fire area should be a strong consideration as PM$_{2.5}$ is predicted to increase as fire area increases

between 0 and 2500 hectares, most steeply between 1500 and 2500 ha, although there is uncertainty at larger fire area because few fires in our data were > 2500 ha (most were < 1000 ha). Note that our fire areas may be an underestimate of total HRB size, as these areas are calculated from VIIRS hotspots, thus is based on active fire area at VIIRS overpass time (2 per day), not total area burnt in a day.

While the models indicate that certain combinations of weather increase PM$_{2.5}$, this must be weighed with the fact that aspects

of HRB implementation cannot always be changed. For example, HRBs are already conducted within the narrow set of weather conditions that allow for ignition and controllable fire spread, and often need to be conducted close to populations to have the greatest house protection effect (Clarke et al., 2019). Due to the complex effects and lower predictive accuracy for HRBs, it is difficult to make precise predictions from the models for individual fires. A more detailed model would be required to identify the weather conditions that would both allow a HRB to be safely conducted and also for PM$_{2.5}$ to be low. An assessment that

combines predictions from our model of lower risk PM$_{2.5}$ days with a model that predicts occurrence of within-prescription HRB burning days (Clarke et al., 2019) may be useful to assess the number of overlapping days, i.e. HRB days with low PM$_{2.5}$ risk. The effects of different burning strategies, such as breaking a large burn up into multiple blocks, are unknown and could potentially worsen PM$_{2.5}$. Here we did not assess different strategies, and only analysed cases where one fire was burning at a time, not when multiple fires were burning around the same AQS at once. This is a significant limitation of the study, as the

smokiest HRB days likely occur when multiple fires are burning at once and/or fires burn for longer periods. Price and Forehead (2021) also suggested overnight burning may have led to the largest PM$_{2.5}$ exceedances that they recorded using low cost monitors near HRBs. Pearce et al. (2012) found burn duration to be important predictor from work also monitoring PM$_{2.5}$ close to HRBs. The effect of total fire load in a region, i.e. total area of all fires, and regional weather conditions is currently the subject of separate research.



**5. Conclusion**

Understanding how individual fires, both wildfires and hazard reduction burns, influence ambient $PM_{2.5}$ concentrations is important to allow for proper risk analysis, burn scheduling and issuance of warnings. Our models provide important insights into the influence of weather and fire variables on $PM_{2.5}$ concentration from individual fires. We found that fire area, PBLH, temperature and RH all have strong influences, with the effects of the variables varying depending on whether it is measured at the fire site or the receiver location (here, the AQS). The models improve our understanding and may have a place during operational predictions. However, accuracy is similar to existing models, so could be used as a complement. Further development to improve accuracy would benefit operational deployment of the models, particularly given the lower correlations between observations and predictions for HRBs. However, our approach is promising and would likely produce better models with a larger set of data, where more cases of single fires near AQS could be found. Increasing the density of $PM_{2.5}$ monitors (permanent or temporary during fire seasons) would also provide better data to improve the resulting models. Producing broader scale models of regional level $PM_{2.5}$ from regional level fire and weather may be a useful next step to produce a predictive $PM_{2.5}$ model for operational use.



## 6. Appendix A

This appendix contains case studies of large $PM_{2.5}$ exceedance events present in the data used for modelling in the main text. The purpose is to detail specific events and highlight factors that may have influenced $PM_{2.5}$ patterns across the different AQS.

The appendix is organised as seven panel figure of seven different events that each have images and a description. The events selected are the six highest mean 6-hour values from the combinations of fire type (WF, HRB) and period (afternoon, evening, morning), and also the second highest value for afternoon WF, which is included to highlight interesting coastal wind behaviour. Note that the values used in modelling are from AQS data for which only one fire was active within 150 km of the AQS for that day. Higher values were recorded on days with multiple fires, but these are not analysed in this paper. Each figure

contains:

- Panel (a) in each figure has a background Himawari 8 satellite image for one single hour (time in black text at top) during the relevant time period, with the fire centroid also indicated by an orange circle and general fire area in blue polygon. The background image is overlaid with wind speed (red numbers and red arrow length) and wind direction (red arrow direction) from Bureau of Meteorology weather stations and $PM_{2.5}$ recorded at all AQS within the image

extent at that hour (black circles and white text, larger $PM_{2.5}$ value means large circle). The AQS with the highest mean six-hour value is indicated by red star (same AQS as general location map). AQS that had multiple fires nearby are not included. Note one extra Himawari image is included for WF night to aid in the description (panel e). Himawari images are provided by Japan Aerospace Exploration Agency (JAXA) and were downloaded the JAXA P-Tree System (https://www.eorc.jaxa.jp/ptree/terms.html).

- Panel (b) in each figure is a map of the general fire location, represented by an orange circle around the fire centroid, with circles representing AQS locations coloured by their mean $PM_{2.5}$ value for that six-hour period. The highest station values are indicated by the red text and red star.
- Panels (c) and (d) in each figure are 10 m and 700 hpa gridded wind speed and direction for the same hour as the Himawari image, sampled from ERA5 gridded reanalysis data. Black arrows indicate wind speed and direction,

with longer/larger arrows indicating higher wind speed. The orange fire circle is also in these images for reference. Black solid line is the Australian coastline.







**Fig. A1: Wildfire afternoon**

The largest afternoon mean WF $PM_{2.5}$ was 294.2 µgm⁻³ at Katoomba AQS, which was the third closest to fire. The fire, Gosper's Mountain fire, eventually burnt ∼ 500,000 ha. 15 other AQS around the fire had mean $PM_{2.5}$ > 50. The smoke was flowing mainly to the south over Katoomba (red star) under the surface northerly winds. However, a portion of the smoke was also flowing more easterly, in line with upper-level winds. There is a distinct easterly edge to the smoke plume (a), which appears to align with where the northerly and north-easterly surface winds meet (c). There were widespread $PM_{2.5}$ exceedances, but only Katoomba recorded values > 150 µgm⁻³ (b). Smoke from other wildfires to the north of Sydney may have affected the region, indicated by the AQS north of the Gosper's fire also recording high PM2.5 values (a). These AQS are not shown in (b) because they were within 150 km of more than one active fire. In the morning before this image, smoke was flowing directly into Sydney and most AQS recorded at least one hourly value > 200 µgm⁻³.



**Fig. A2: Wildfire afternoon 2**

The second highest mean afternoon WF PM$_{2.5}$ was 138.1 μgm$^{-3}$ at Richmond AQS (red star), which was the closest to fire at ~50 km. This was also the Gosper's Mountain fire. Six other AQS in the Sydney basin had mean PM$_{2.5}$ > 50 (b). The smoke flow appears multidirectional (a): to the south and to the east, more in line with upper winds (d). The smoke flowing to the east appears to be above the surface and above the meeting of westerly winds at 10 m and E/NE sea breeze at 10 m (c). The smoke flowing to the east appears to be above the surface and above the meeting of the sea breeze as the AQS directly to the east of the fire was under the plume but only recording PM$_{2.5}$ of 7 μgm$^{-3}$ at this hour (a). The morning preceding had northerly winds blowing smoke directly into Sydney, with several AQS recording hourly values in the 100s, so lingering smoke was likely a contributor. This fire burnt through the following night, but PM$_{2.5}$ levels dropped substantially across Sydney (all < 25 μgm$^{-3}$) until 2am, when PM$_{2.5}$ increased again (Fig. A4).





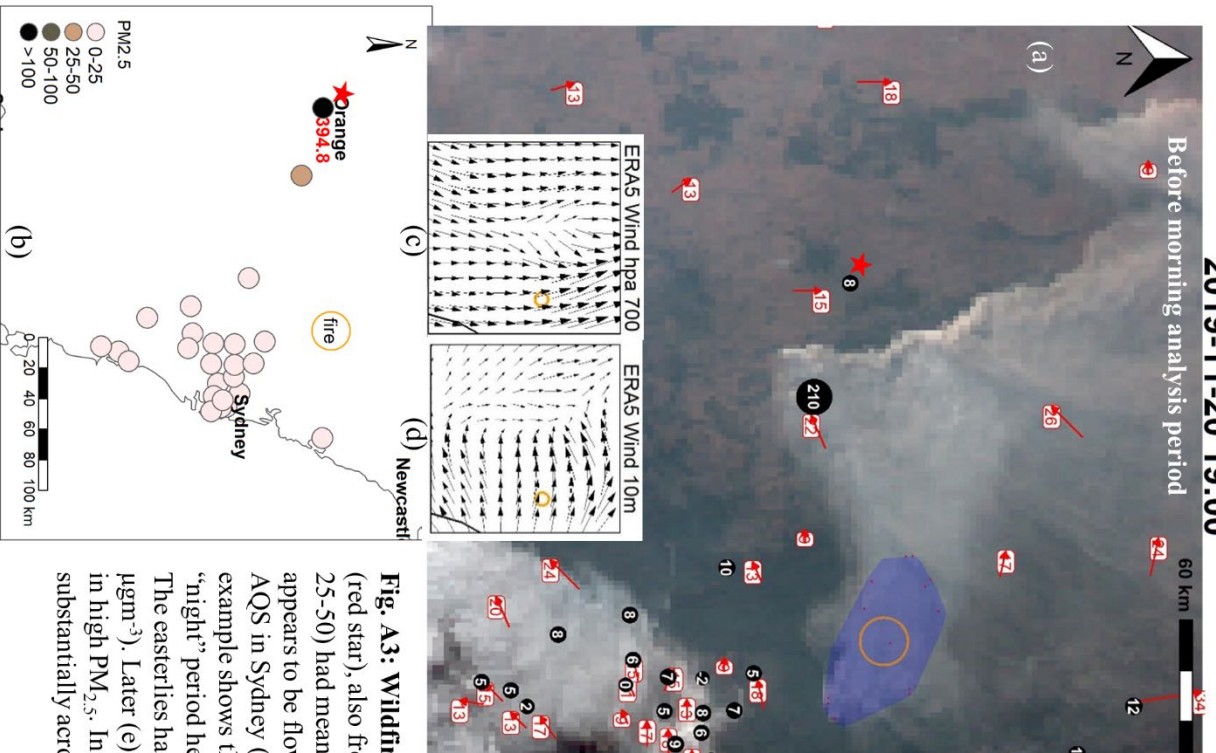

**Fig. A3: Wildfire night.** The highest mean night WF $PM_{2.5}$ was 394.8 $\mu gm^{-3}$ at Orange (red star), also from the Gosper's Mountain fire. All except one other AQS (Bathurst 25-50) had mean $PM_{2.5} < 25$ $\mu gm^{-3}$ for this period (b). At midnight (e) the smoke appears to be flowing directly from the fire to Orange ($PM2.5 = 462$ $\mu gm^{-3}$), given no AQS in Sydney (SE of fire) had high $PM_{2.5}$, and the easterly 10 m winds (f). This example shows that smoke circulation can be complex. At 19:00, one hour before our "night" period here, there were easterlies at the fire but Orange AQS had southerlies (a). The easterlies had only reached Bathurst ($PM_{2.5} = 210$ $\mu gm^{-3}$) and not Orange ($PM_{2.5} = 8$ $\mu gm^{-3}$). Later (e) the easterlies reached Orange along with the surface smoke, resulting in high $PM_{2.5}$. In the day after this, winds switched to northerlies and $PM_{2.5}$ rose substantially across Sydney and in Katoomba.




**Fig. A4: Wildfire morning**

The highest mean morning WF PM$_{2.5}$ was 506.2 $\mu$gm$^{-3}$ at Rouse Hill (red star), which was the second closest AQS and 30-40 km from the fire edge. This was also the Gosper's Mountain fire. Richmond AQS was closer but recorded mean PM$_{2.5}$ of 112 $\mu$gm$^{-3}$. There were seven other AQS with a mean > 100 $\mu$gm$^{-3}$ for this period and another three > 50 $\mu$gm$^{-3}$ (b). It is not apparent from the BOM wind (red arrows in a) or the ERA5 data (c and d) that wind flowed directly from the fire to Rouse Hill. The high values may be the result of smoke lingering from the previous day (Fig. A2). However, smoke appeared to clear out during the night period preceding this morning, as the night had comparatively low PM$_{2.5}$ values (all Sydney AQS < 25 $\mu$gm$^{-3}$). There may have been drainage flows from the mountains into the Sydney basin that were not captured in the weather data. After this morning, winds turned westerly then southerly, which pushed smoke away from Sydney and reduced PM$_{2.5}$ < 25 $\mu$gm$^{-3}$ across Sydney. Smoke passed over the AQS in the NE of the image (Newcastle), however, PM$_{2.5}$ values remained relatively low, suggesting the plume was above the surface.



**Fig. A5: HRB afternoon**

The highest mean afternoon HRB $PM_{2.5}$ was 145.9 $\mu g m^{-3}$ at the closest AQS in Richmond (red star). Four other AQS also recorded $PM_{2.5} > 50$ $\mu g m^{-3}$ during the same period (b). Wind patterns at 10 m (c) suggest generally northerly winds, and the meeting of NW and NE winds west of the fire that possibly funnelled surface smoke to Sydney, similar to Fig. A2. Wind speeds were also low (a and c), meaning smoke would have remained in the area for longer. On the morning preceding this period, there was likely some smoke lingering around Sydney because several AQS recorded hourly values in the 20s and 30s ($\mu g m^{-3}$). The synoptic wind direction (d) in the afternoon was northerly, meaning high-level smoke would also have flowed toward Sydney. $PM_{2.5}$ continued to increase as the day went on and into the night. The highest night mean $PM_{2.5}$ was also caused by this HRB (Fig. A6).

2018-05-28 16:00

(a)

Base image courtesy of JAXA

60 km

PM2.5
○ 0-25
● 25-50
● 50-100
● >100

ERA5 Wind 10m

(c)

ERA5 Wind hpa 700

(d)

(b)

N

145.9

fire

Sydney

Newcastle

0 10 20 30 40 50 km





**Fig. A6: HRB night**

The highest mean night HRB PM$_{2.5}$ was 161.7 µgm$^{-3}$ at the second closest AQS in St Marys (red star), while the closest station was also high (Richmond = 129.4 µgm$^{-3}$). This highest night-time PM$_{2.5}$ for HRBs followed the highest afternoon PM$_{2.5}$, i.e. same day and fire (Fig. A5). The high afternoon PM$_{2.5}$ may have influenced the results this night as 10 m winds were still low (c), suggesting smoke may have lingered from the afternoon. Low wind speeds also meant that any new smoke produced during the night was probably not dispersed. The 10 m winds (c) suggest smoke was not flowing directly from the fire to St Marys, as winds were westerly at the fire. However, the BOM winds (red arrows in a) are zero at Richmond and St Marys at 22:00, suggesting very calm conditions.





**Fig. A7: HRB morning**

The highest mean morning HRB PM$_{2.5}$ was 284.5 µgm$^{-3}$ at Bargo AQS (red star), which was only 10 km from the HRB. This example differs to the other panels in that PM$_{2.5}$ impact was local only: no other AQS were > 25 µgm$^{-3}$ for this morning (b). None of the examples (Fig. A1-A6) had AQS at such a close distance. The 10 m winds (c) suggest light wind from the fire toward the AQS. The BOM winds near the fire (a) varied in direction but were light. The wind speeds were likely too low to carry smoke far enough north to impact AQS in Sydney, or AQS in other directions. This HRB was ignited the previous day under W/SW winds, but did not noticeably increase PM$_{2.5}$ at any AQS. In the day after this morning period, strong northerly winds occurred and PM$_{2.5}$ at Bargo dropped below 10 µgm$^{-3}$.



## 7. Author Contributions

Owen Price developed the research aims. Michael Storey and Owen Price developed the analysis method. Michael Storey ran the statistical analysis and wrote the manuscript. Owen Price edited the manuscript.

## 8. Data Availability

VIIRS SNPP hotspots used for analysis are freely available via the NANA FIRMS website at https://firms.modaps.eosdis.nasa.gov/download/

New South Wales $PM_{2.5}$ data are freely available from the New South Wales Government at free online at https://data.airquality.nsw.gov.au/docs/index.html.

Information on ERA 5 gridded reanalysis weather product download is at https://www.ecmwf.int/en/forecasts/datasets/reanalysis-datasets/era5

## 9. Acknowledgements

This research was funded by the NSW Department of Planning, Industry and Environment, via the NSW Bushfire Risk Management Research Hub.

The results contain modified Copernicus Climate Change Service information 2021. Neither the European Commission nor ECMWF is responsible for any use that may be made of the Copernicus information or data it contains.

## 10. Competing interests

The authors declare that they have no conflict of interest.

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
