# Peer review of "Statistical modelling of air quality impacts from individual forest fires in New South Wales, Australia"

_EGUsphere, 2022_

## Author Response (AR1)

**Author responses to referee commments on egusphere-2022-345.**

We thank the reviewers and editor for reviewing our manuscript and appreciate the comments and suggestions. We have taken a detailed look at all the comments and made appropriate changes.

We would like to highlight we were required to change our method of fire area calculation. We originally calculated fire area based on the number of VIIRS hotspots. However, we found that this overestimated area because VIIRS hotspots can overlap from different acquisitions, ie the same place could be counted twice. We revised our are calculation method to now be one based on the intersection of all VIIRS hotspots per day with a 500 m by 500 m set grid. This change in method resulted in only minor changes to the results, plots and interpretation of results. We have updated our plots and results table (models and accuracy stats are slightly different) to reflect this change. The VIIRS area method is described in section 2.1.

Below are the referee comments in black and our responses in red.

Please see the "track changes" manuscript copy for further detail if required.

**Comment on egusphere-2022-345**

Anonymous Referee #1
* * *
Referee comment on "Statistical modelling of air quality impacts from 1500 individual forest fires in NSW, Australia" by Michael A. Storey and Owen F. Price, EGUsphere, https://doi.org/10.5194/egusphere-2022-345-RC1, 2022
* * *
This paper models PM2.5 emitted from 1500 historical individual fires in NSW Australia, as a function of fire and weather variables. The PM2.5 values were measured at air quality stations in NSW Australia. The study combines data from satellite, reanalyses of meteorological data and monitoring station's data (for air quality). VIIRS satellite hotspots were used to identified days where one fire was burning within 150 km of one of 48 air quality station. To model and predict air quality events, the authors used random forest models for afternoon, night and morning PM2.5.

The authors identified the most important variables and drivers which are a mix between fire characteristics and meteorological conditions, namely fire area, boundary layer height, temperature, wind speed and relative humidity. The models underestimate the observed pollution values.

The subject discussed in the present article is of high importance, and the paper is well structured and easy to follow. The introduction is very clear and includes interesting and recent references. The objectives are very well presented. The limitations of the study are in a great measure identified. This is an important point, as one of the caveats relates to the fact that not all fires are accounted and some of the fires not accounted (because of the co-occurrence of fires) can exclude fires which contributed greatly to large pollution events. The discussion is quite interesting. Still, there are points that need further details and work. I believe that this document should be considered for publication after minor/moderate changes.

Below I point some comments and suggestions, which hopefully can help the authors to enhance the manuscript.

Comments:

- **Introduction:** The intro starts citing a reference which is 10 years old. Is this true now? Please comment on or change it to reflect more recent numbers.

  We added some other more recent references here

- **Introduction: Minor comment, Line 94,** "We need to better tools (…)"

We corrected this

- **Introduction: Line 96-99,** Add some references to justify on why these attributes are important.

  We have added supporting references

- **Introduction:** Another alternative to the methods cited is using satellite information

  Please add some information on that, e.g., Gupta et al., 2006
  https://www.tandfonline.com/doi/full/10.1080/01431160701241738
  We now mention this and provide a reference

- **Introduction:** The option on using random forest models is not justified in any way in the introduction. Some information should be added on this type of approach. Another kind of modelling approaches could be used and should be mentioned, e.g., neural networks, regression models, GAM, among others. Why are random forest models better than other options? Please justify.

  We have added a couple of sentences about random forest modelling and its advantages, which we believe is best placed in the methods.

- **Methods: Line 158,** Why use a threshold of 125? Please justify

  This is actually a scaled score and translates to a cover fraction of 0.25. We updated the manuscript to say 0.25 instead as it makes more sense. Our study focused on forest fires. This threshold ensured only dense woodlands, open forest and closed forest was included in our study.

- **Methods: Line 159-160**, Why use a buffer of 2.5 km? Please justify.

  We provided some further information here, and acknowledge that using a different buffer may have produced slightly different results (see Limitations section 2.5).

- **Methods: Line 164-166,** The results for the 3-days window could have been shown in the sup. material.

  Would believe our mention of this is sufficient here and that added a second set of plots in the appendix would not be all that informative. Given this was not raised by the other reviewer, we would prefer not to make any changes here.

- **Methods: Fig.1 and Line 203,** How do you justify using monitoring stations with data records with only 3 or less years of records? In what way the exclusion of stations with shorter lengths would influence the results of the aggregation process?

  For a new row in our data tables, for each active fire we only required at least one day (afternoon, night & morning periods) at an air quality station to be captured. We did not make the sampling dependent on the length a station is active, we only searched for occasions where any active fire was near any available station. Excluding short term records may alter results (probably not change the overall conclusions much), but much of our data does come from more recent years due to many fires in this period and more stations becoming active.

- **Methods: Line 211**: "We also calculated the sum of the hotspot day and night fire area as a predictor". The sum of what? Please clarify.

  Note that we had to change our method of area calculation (which produced very similar final results), so this section will be updated in a revised manuscript.

- **Methods: Section 2.4,** Please add some more information on random forest models, namely, way to apply, advantages and caveats.

  We added a few sentences on random forests here, as also suggested in a prior comment.

- **Methods:** Why do you opt by doing a simple validation (75% training and 25% test) in detriment of a cross-validation or other validation method. Please justify.

  We were aware of several approaches to cross-validation that could possibly be used. The random forest out-of-bag r-squared is an in-built independent validation method which we have reported on for our training steps. In the end, and given that we had

sufficient data, we chose to use a simple training-test split which is a regularly used approach. Adding results from a completely independent set has the advantage of giving a practical example of accuracy of predictions to a new test set and confirming the OOB accuracy. The prediction observation on the training (OOB) sets are consistent with the correlation on the test sets, which suggests changing the validation approach wouldn't change the results substantially.

Do you compare models' outputs to observations in an independent sample, not used to create the models? An independent sample was mentioned in the results section but information on this should be added on the methods.

Yes, the comparison with the test set is included in Table 1 and Figure 7. We refer to the table and figure in the results and state the accuracy on the independent test set too.

- **Results:** Did you analyse if the under/over predictions were due to models not being able to capture the intensity of the events or because they are able to capture that but with a delay? This is important to try to understand and correct models' performance.

  We did not look at this detail. We could have a further look. It would be helpful if a couple of references could be provided that look at this issue in other random forest models.

- **Results:** Table 1: Please verify the legend.

  We have updated this to "Table 2", as it was mis-labelled. The rest if the caption looks accurate.

- **Results:** Do you think it might be possible that night results are worse as afternoon presents higher values and thus the models have a harder time to reproduce night values? If this is true, how can you correct it?

  The distribution of $PM_{2.5}$ for night and afternoon are fairly similar: Vast majority are < 20 $ugm^{-3}$ for morning and night, with a small proportion > 20 (Figure 3). The 20 highest values for each (for both WF and HRB) are also within a similar range, being mostly 50 and 150 $ugm^{-3}$. There is also only a small difference in accuracy on the training and tests sets in terms of prediction and observation correlation (~0.8). Given this, we don't believe differences between the distributions of $PM_{2.5}$ between morning and night would have greatly affected the model results.

- **Discussion:** One of the caveats of the approach is that only considers part of the fires. Therefore, the first goal is only partially achieved as a great part of the fires and corresponding weather conditions are not analysed.

  We have now ensured that in the methods, results and discussion that fire area is referred to as "daily active fire area", so that it clear we are not referring to the total final fire area. We believe this is the clarification needed, but if there is a further issue needing attention, please provide some more details and we are happy to look at it.

- **Discussion:** The authors do not account nor mention the effects of recirculation potential on these events, nor even the link to PBLH. There several papers connecting poor air quality events and recirculation and would be nice to refer them in the discussion.

https://www.sciencedirect.com/science/article/pii/S1352231002009263

https://www.sciencedirect.com/science/article/pii/S1309104221003305

https://agupubs.onlinelibrary.wiley.com/doi/pdf/10.1029/2007JD009529

Thanks for the references. We have added recirculation into an existing part in the discussion about broader weather patterns, and included these references. We have briefly discussed PBLH in the discussion and believe this is sufficient, particularly given the discussion is already quite long.

EGUsphere, referee comment RC2
https://doi.org/10.5194/egusphere-2022-345-RC2, 2022
**Comment on egusphere-2022-345**

Anonymous Referee #2
* * *
Referee comment on "Statistical modelling of air quality impacts from 1500 individual forest fires in NSW, Australia" by Michael A. Storey and Owen F. Price, EGUsphere, https://doi.org/10.5194/egusphere-2022-345-RC2, 2022
* * *
The study modeled PM2.5, measured at air quality stations in NSW Australia, from 1500 historical individual fires as a function of fire and weather variables. It combines fire satellite data, air quality data and ERA5 gridded weather data, and forest models to predict PM2.5 concentrations. The models provided the identification of the main drivers of PM2.5 from individual fires, including fire area and meteorological parameters such as Planetary boundary Layer Height, temperature, wind speed, and relative humidity. The results are important for a better understanding of the impacts of individual fires, both wildfires and hazard reduction burns, on ambient PM2.5 concentrations, and the influence of fire and weather conditions. The manuscript is overall well written, and I have a few comments and suggestions:

**Introduction: Line 85**, I don't know if it's clear to a reader from another country that NSW is a state of Australia, I suggest presenting some information regarding it here. We have added some extra information for clarity.

**Introduction: Line 97**, "proximity to human populations" does not affect "PM2.5 output", as the phrase suggests. Proximity affects the population exposure, I suggest rephrasing.

We have now updated this to clarify our meaning

**Introduction:** I missed a brief discussion on using random forest models. Are there previous studies using this approach? Why are these models suitable for this work? There are limitations?

We have added a few sentences about random forest modelling and its advantages, which we believe would be best placed in the methods.

**Methods - Fire Data, Line 155-167**, The choice of buffers and thresholds seemed a bit arbitrary. Was there some sensitivity test? Any justification for choosing these values (distance of 150 km, foliage projective cover>125, buffer by 2.5 km)?

As for other reviewer: This is actually a scaled score and translates to a cover fraction of 0.25. We will update the manuscript to say 0.25 instead as it makes more sense. Our study focused on forest fires. This threshold ensured only dense woodlands, open forest and closed forest was included in our study.

We can provide some further information about the 2.5 km buffer in an updated manuscript, and acknowledge that using a different buffer may have produced slightly different results (see Limitation section 2.5).

150 km captures most of the eucalypt dominated Blue Mountains that is subject to the majority of fire activity near Sydney. We have added this detail into a revised manuscript.

**Methods - PM2.5 Data, Line 191,** The PM2.5 definition ("particulate matter < 2.5 μm diameter as micrograms per cubic meter of air") could have been presented at the beginning of the manuscript.

We have now included this information in the introduction

**Methods - PM2.5 Data**: The authors do not mention any issues related to data validation or missing data. This happened? If so, what was the strategy to resolve this?

We have add a couple of sentences about NA values for PM$_{2.5}$ data. There were no issues with ERA5 of VIIRS hotspots data because these were downloaded in a standardized/pre-processed format. There were some accuracy issues in the fire history polygon data, which we have mention in section 2.1.

**Methods, Line 212:** Please check the writing: "modelling to account for account for seasonal"

We agree this is unclear, and have updated the sentence so the meaning is clearer.

**Results, Figure 3 -** I didn't see the unit of measurement on the x-axis of the first graph (PM2.5). Is this ugm-3?

We update the plot with the correct units ($\mu gm^{-3}$)

**Discussion,** In general, when analyzing the influence of fire conditions and weather conditions on PM2.5 dispersion, the work does not consider the role of the formation of secondary aerosols from directly emitted precursor gases throughout plume transport. Can secondary PM loading (from smoke precursors) under- or over-estimate the modeled PM2.5? Which predictor variables might have a greater relationship with this effect? It would be nice to include some discussion on this in the manuscript.

We suggest the best solution would be to add a few sentences in the methods that say that observed PM2.5 at the air quality stations could be primary or secondary, which can be influenced by sunlight, temperature, humidity etc., but our analysis did not distinguish between primary and secondary. These sentences have been added in the methods.